# Sleep Disorders, Dysregulation of Circadian Rhythms, and Fatigue After Craniopharyngioma—A Narrative Review

**DOI:** 10.3390/biomedicines13102356

**Published:** 2025-09-26

**Authors:** Hermann L. Müller

**Affiliations:** Department of Pediatrics and Pediatric Hematology/Oncology, University Children’s Hospital, Klinikum Oldenburg AöR, Carl von Ossietzky Universität Oldenburg, Oldenburg 26133, Germany; hermann.mueller@ymail.com

**Keywords:** craniopharyngioma, obesity, sleep, fatigue, circadian rhythm, quality of life, hypothalamus

## Abstract

**Introduction**: Tumor- and/or treatment-associated hypothalamic damage results in reduced quality of life and increased morbidity due to sleep disorders in survivors of craniopharyngioma. **Methods**: The narrative review is based on a search of Web of Science, MEDLINE/PubMed, and Embase databases for the identification of publications. The search terms craniopharyngioma, sleep disorders, fatigue, and daytime sleepiness were used. Selected English language papers published 1970–2025 were included. **Results**: Circadian rhythms (wakefulness and sleep) are controlled by hypothalamic suprachiasmatic nuclei and regulated by melatonin. A dysregulation of circadian rhythms due to altered melatonin secretion can be observed in craniopharyngioma with hypothalamic involvement. Furthermore, sleep quality is regulated by lateral hypothalamic areas, the ventrolateral preoptic nucleus, and monoaminergic nuclei which function as the arousal system. Flexible changes between sleep and wakefulness can be achieved through interaction of arousal and sleep-promoting systems named “flip–flop” switch. Insomnia can be the result of damage to the ventrolateral preoptic nucleus. Excessive daytime sleepiness and disrupted sleep patterns can be observed due to dysregulation of lateral hypothalamic areas. Obesity, chronic fatigue, headache, and excessive daytime sleepiness can be the result of poor sleep quality. “Primary” hypothalamic sleep dysfunction, including narcolepsy, dysregulated sleep–wake cycles, and hypersomnia, can be observed due to hypothalamic dysfunction. “Secondary” sleep disturbances including obstructive sleep apnea, insufficient substitution medication for arginine vasopressin deficiency (nocturia), or psychosocial factors are sequelae in patients with craniopharyngioma and hypothalamic lesions. **Conclusions**: Further research on novel treatment approaches for sleep disorders due to hypothalamic syndrome are warranted to improve the outcome after craniopharyngioma.

## 1. Introduction

Craniopharyngioma (CP) [1] is a rare embryonic malformation located in the sellar/parasellar area of the skull base [2,3,4,5]. Craniopharyngiomas are differentiated into adamantinomatous and papillary types by the updated world health organization (WHO) classification of central nervous system tumors published in 2019 [6,7]. Adamantinomatous CPs are primarily diagnosed in the pediatric age group, whereas papillary CPs mostly affect adults. The overall incidence of CP is low, with only 0.5–2 newly diagnosed cases per 1,000,000 people per year [2]. For the adamantinomatous type, a median age peak at CP diagnosis ranges from 5 to 9 years and from 55 to 69 years for the papillary type. However, CP can be diagnosed at an early age including neonatal period and infancy. Even prenatal diagnoses have been reported [2,8,9,10]. The survival rates after CP range from 90% to 96% and have increased during the past [2,11].

Due to the close neighborhood between the CP and hypothalamic nuclei, optic nerves, the optic chiasm, and the pituitary gland, patients with childhood-onset CP frequently suffer from (neuro) endocrine deficits [2,11,12,13,14,15], weight gain and morbid hypothalamic obesity [16,17,18], severe impairments of health-related quality of survival [13,19,20,21,22,23,24], impaired visual function [25], and cardiovascular and metabolic morbidity [26,27,28,29]. Disturbances of circadian rhythms and fatigue are major complaints of CP patients affected with hypothalamic syndrome [18,30,31,32]. Their treatment causes considerable costs [10,33].

Many pediatric patients with childhood-onset CP also have sleep disorders including impaired melatonin secretion and secondary narcolepsy [34,35,36,37,38,39,40], associated with hypothalamic syndrome [41]. Significant impairments in daily levels of activity and social function are clinical manifestations [42,43,44,45,46,47]. Fatigue is another major symptom associated with hypothalamic syndrome and (neuro) endocrine deficiencies [48]. Fatigue as a multidimensional clinical symptom has an effect on cognitive, physical, or emotional domains in daily life [49]. Fatigue reduces quality of life after childhood-onset CP [20,50,51]. Patients with CP and hypothalamic syndrome also suffer from excessive daytime sleepiness, as quantified by Epworth Sleepiness Scale (ESS) [52]. The clinical manifestation of fatigue is characterized by decreased physical and mental ability and increased tiredness. Patients with these symptoms indicating fatigue do not experience rest through sleep. [53]. Hypothalamus-sparing surgical and radio-oncological strategies can help to prevent hypothalamic syndrome, which is associated with fatigue [54,55].

## 2. Search Strategy

This narrative review aims to assess the occurrence and severity of sleep disorders such as fatigue and daytime sleepiness in CP patients, with and without hypothalamic syndrome, to analyze the symptoms of sleep disturbances, and to analyze risk factors for disorders of sleep and circadian rhythms.

The narrative review is based on a literature search of Web of Science, MEDLINE/PubMed, and Embase databases for initial identification of publications. The articles were identified using the following keywords: a) “craniopharyngioma” and b) “sleep,” “sleep disorders,” “sleep-related breathing disorders,” “sleep-disordered breathing,” “excessive daytime sleepiness,” “hypersomnolence,” “narcolepsy,” “fatigue,” “circadian rhythm,” “melatonin,” “stimulant.” Selected English language papers published between 1970 and February 2025 were included in our review. Only studies published in peer-reviewed journals were considered.The inclusion criteria, based on the PICOS approach, were as follows: participants (P): patients diagnosed with craniopharyngioma; intervention (I): no restrictions; comparison (C): no restrictions; outcomes (O): sleep-related parameters; study design (S): original studies. Exclusion criteria included: (1) non-English language publications; (2) animal studies or in vitro research; (3) studies involving participants with craniopharyngioma with additional central nervous system complications; (4) reviews, case report, thesis, and conference proceedings. In cases where multiple studies were published based on the same databases, only the study with the largest sample size was included in the analysis. Thirty-eight publications fulfilled the above-mentioned criteria (Figure 1).The data extracted from all studies in line with our research objectives included participants characteristics (i.e., age of the subjects, age of tumor onset, clinical characteristics), procedures used to evaluate sleep disorders (subjective and/or objective investigations), type of sleep disturbance, any additional examinations (melatonin dosage), treatment and efficacy, if available.

## 3. Results

### 3.1. Pathophysiology of Sleep Disorders in Craniopharyngioma

Major regulators of circadian rhythm with significant clinical effects on wakefulness and sleep are the suprachiasmatic nuclei. Melatonin secreted during darkness by the pineal gland has a major influence on circadian rhythms [56,57,58]. Due to alterations in hypothalamic–pituitary function, dysregulation of melatonin secretion can be found in CP patients. In CP patients, reduced nighttime melatonin secretion and high cortisol serum levels are related to reduced total sleep duration, daytime physical activity, sleep efficiency, time of sleep, and higher frequency of awakening [59]. Nighttime and morning melatonin concentrations in saliva show a negative correlation with body mass index (BMI) [60] and daytime sleepiness [35]. Important regions for sleep regulation are the suprachiasmatic and the median preoptic nuclei, and the ventrolateral preoptic area [61]. These hypothalamic areas contain g-aminobutyric acidergic neurons. By the activation of these neurons, a promotion of sleep is achieved via descending projections on the posterior and lateral hypothalamus, and via the brainstem [62]. Arousal pathways lead to the direct stimulation of the cerebral cortex via orexin-expressing neurons located in the lateral hypothalamus [63]. These arousal and sleep-promoting pathways have mutual effects on each other by a physiological mechanism called the “flip–flop” switch [64]. This “flip–flop” switch regulates the changes between waking and sleeping. Damage of nuclei in the preoptic hypothalamic region produces insomnia, and damage of the lateral hypothalamic area results in hyposomnia [65,66].

Narcolepsy is a clinical syndrome including excessive daytime sleepiness and episodes of cataplexy, sometimes combined with hallucinations and sleep paralysis. In a subgroup of patients with narcolepsy, reduced orexin concentrations resulted in a destabilization of the flip–flop switch [67,68]. Only a few publications, including mainly case reports, observed secondary narcolepsy following brain tumors [69]. Khan et al. reported on a prevalence rate of 1670 per 100,000 for narcolepsy/hypersomnia in survivors of pediatric central nervous system (CNS) tumors [70]. However, this reported rate most likely underestimated the prevalence because a systematic sleep assessment was not performed in all CNS tumor survivors. Midline CNS tumors, irradiation doses > 30 Gray (Gy), and epilepsy medication were factors associated with a higher risk for narcolepsy and hypersomnia. The high prevalence rate of excessive daytime sleepiness in up to 30% of patients with childhood-onset CP could explain the higher rate of hypothalamic obesity by the reduction of daily physical activity [35,71,72].

### 3.2. Assessment of Sleep Disturbances

“Primary” hypothalamic dysfunction as for instance disturbed cycles of wake/sleep, narcolepsy, and hypersomnia can cause sleep disturbances in CP patients [73,74]. “Secondary” causes of sleep disturbances are hypothalamic obesity and consecutive obstructive sleep apnea syndrome. Diagnostics of sleep disturbances must also consider the perspectives of family and partners. The initial diagnostics of sleep disturbances should include the use of a sleep diary to assess sleep characteristics for at least one week [75,76]. Additionally, physical movements during day and night can be monitored by an actigraphy device. The degree of daytime sleepiness can be assessed by ESS, a specific and validated questionnaire for daytime sleepiness [52]. The multiple sleep latency test (MSLT) provides a more precise measurement of narcolepsy, hypersomnia, and daytime sleepiness [77]. Secondary narcolepsy in patients with childhood-onset CP may be similar to idiopathic forms of narcolepsy, characterized by high frequency of sleep-onset rapid eye movement periods (SOREMPs) in MSLT and low orexin concentrations in cerebrospinal fluid (CSF) [34,78,79,80]. Obstructive sleep apnea syndrome or disruptions of the sleep–wake cycle should be diagnosed and monitored by polysomnography [81]. Melatonin and cortisol are circadian phase markers with close relation to sleeping patterns and BMI in patients with childhood-onset CP [36] (Figure 2). However, these parameters are not suitable for routine diagnostics of sleep–wake cycle disturbances [35,59,82].

### 3.3. Excessive Daytime Sleepiness

Already in 1970, Killeffer et al. published the case of a pediatric CP patient suffering from disturbances of sleep patterns after a surgical intervention, frequent episodes of falling asleep, disturbances of circadian rhythms, reversal of day-to-night sleep rhythm, and excessive daytime sleepiness [83]. Considering all reports regardless of diagnostic techniques used, excessive daytime sleepiness is recognized with a rate ranging from 25% to 100% of cases. Different cohort sizes and criteria for patient selection could explain the wide range of variability. Lower prevalence values for somnolence—ranging from 25% to 43% of cases—have been observed in analyses questionnaires. Poretti et al. observed excessive daytime sleepiness in 29% of patients with childhood-onset CP [84]. In an ESS study on seventy-nine CP patients, Müller et al. reported excessive daytime sleepiness in 42% of severely obese and in 35% of less obese or normal weight CP patients [35]. Van der Klaauw et al. observed hypersomnolence in 33% of CP patients [85]. The authors compared healthy controls with CP patients and patients with non-functioning pituitary macroadenomas. They observed that patients with CP or non-functioning pituitary macroadenomas have excessive daytime sleepiness despite normal nocturnal sleep patterns [85]. Manley et al. observed excessive daytime sleepiness in 43% of patients with childhood-onset CP. [86]. An excessive daytime sleepiness (ESS values > 10) prevalence rate of 25% was reported by Crowley et al. [87]. Using polysomnography and MSLT as the standard for objective assessment of excessive daytime sleepiness, the prevalence rate of excessive daytime sleepiness in CP patients appears to be higher, ranging from 50% to 100% of cases. Based on mean sleep latency values < 10 min in MSLT, Crabtree et al. found that 82% of pediatric CP patents displayed excessive daytime sleepiness [88]. Using a modified ESS, the authors observed excessive daytime sleepiness in 29% of the patients. The results lead to the speculation that CP patients often do not correctly report or recognize their sleep disorders. In polysomnography and MSLT after surgical resection of CP, Mandrell et al. observed excessive daytime sleepiness in 80% of patients with CP [80].

In a study using sleep questionnaires, MSLT, and polysomnography, Pickering et al. found that patients (7 CP) felt sleepy more frequently than controls (*n* = 10), and that 57% of CP patients presented with electrophysiological signs indicative of hypersomnia on MSLT [89]. Snow et al. reported on an increase in subjective daytime sleepiness (ESS: 15.2 ± 2.8 in tumor patients; 5.00 ± 2.00 in controls) and objective daytime sleepiness in five patients with pituitary tumors (mean MSLT sleep latency: 10.3 ± 5.3 min in patients; 26.2 ± 1.1 min in controls) [90]. Using actigraphy and polysomnography, Niel et al. found a prevalence rate for hypersomnia of 50% in pediatric patients with childhood-onset CP and an age ranging from 3 to 20 years [91]. Using MSLT, the authors observed excessive daytime sleepiness in 82% of seventy-eight CP patients [47]. Using MSLT, polysomnography, and a modified version of the ESS for sleep evaluation in sixty-two pediatric CP patients, Jacola et al. reported that 76% of the analyzed group met the MSLT-based diagnostic criteria for excessive daytime sleepiness. The degree of excessive daytime sleepiness was positively associated with the grade of hypothalamic involvement [92]. Yang et al. analyzed the associations between sleepiness as measured by ESS and grade of hypothalamic damage in one hundred thirty-one CP patients. The authors found that sleepiness was worse in the CP group with bilateral hypothalamic involvement when compared with the subgroups of CP patients with unilateral hypothalamic involvement, mild hypothalamic involvement, or without hypothalamic involvement [93]. Analyzing eighty-four pediatric CP patients, Klages et al. observed a positive correlation of hypothalamic involvement of CP with BMI and excessive daytime sleepiness [50].

Using the Pediatric Quality of Life Questionnaire (PEDQOL), Mann-Markutzyk et al. [94] analyzed the relation between health-related quality of life and excessive daytime sleepiness in CP patients. CP patients with excessive daytime sleepiness (ESS score > 10, *n* = 34) presented with an impaired self-assessed quality of life (QoL) (*p* = 0.003), compared to CP patients without excessive daytime sleepiness (ESS < 10; *n* = 85). Daytime sleepiness as quantified by ESS showed a negative correlation with the quality of life (r = −0.395; *p* < 0.001). Surgical hypothalamic damage, observed in 93% of the reference-assessed postsurgical magnetic resonance imaging (MRI), resulted in increased ESS and excessive daytime sleepiness, whereas such an association was not detectable for presurgical hypothalamic involvement of CP (observed in reference-assessed presurgical MRIs of 72% of CP patients).

Fatigue as a sequela of previous tumor treatment, inappropriate timing and/or dosage of endocrine substitution medication for pituitary deficiencies, and various psychosocial disorders may have additional effects leading to reduced quality of sleep and increased daytime sleepiness [95]. The effect of irradiation and various radio-oncological techniques—such as proton beam therapy vs. photon-based irradiation—on the development of fatigue after CP treatment needs further research [96,97,98,99,100,101].

### 3.4. Secondary Narcolepsy

Secondary narcolepsy can be a potential disease causing excessive daytime sleepiness in patients with childhood-onset CP. Of all the cases reporting on excessive daytime sleepiness, only a few reports show data on narcolepsy, observing a prevalence rate of secondary narcolepsy ranging from 14% to 35% [50,80,89,91]. Obese and overweight CP patients presented more frequently with narcolepsy or hypersomnia. At the time of CP diagnosis, clinical signs of secondary narcolepsy were detected more frequently in patients with grade 2 hypothalamic involvement of CP (anterior plus posterior hypothalamic involvement, including the mammillary bodies) [80]. In a retrospective analysis on secondary narcolepsy, Madan et al. found the narcolepsy type in three of ten CP patients [102]. Marcus et al. reported on a pediatric patient with childhood-onset CP and secondary narcolepsy without the clinical manifestation of cataplexy. In this patient, excessive daytime sleepiness started 4 weeks before the time of CP diagnosis [79]. Patients suffering from narcolepsy type 1 display a dysregulation of wake boundaries and sleep (flip–flop switch) due to a lack of the wake-promoting effects of the neuropeptide hypocretin [64]. The current diagnostic criteria for narcolepsy type 1 are as follows: a mean MSLT sleep latency < 8 min, ≥ 2 sleep-onset rapid eye movements periods (SOREMPs) on MSLT, and decreased hypocretin-1 (hcrt1) cerebrospinal fluid (CSF) concentrations or clear-cut cataplexy [103]. In CP patients with narcolepsy, reduced CSF concentrations of hcrt-1 [68,104] have been found, regardless of whether cataplexy was detectable. These findings suggest that surgical interventions affecting hypothalamic nuclei could result in impaired orexin secretion and excessive daytime sleepiness [78,105]. However, Pickering et al. and Snow et al. found normal CSF concentrations of hypocretin [89,90]. However, it should be noted that Snow et al. analyzed CP patients with excessive daytime sleepiness, who did not suffer from [90]. Similarly, Pickering et al. measured hypocretin in CP patients, but not in the subgroup of patients characterized by electrophysiological criteria of narcolepsy [89]. However, it is not clear whether hypocretin CSF concentrations were analyzed in childhood-onset CP patients.

### 3.5. Sleep-Disordered Breathing/Obstructive Sleep Apnea

Sleep-disordered breathing is one of the causes of excessive daytime sleepiness. However, studies analyzing a potential association between obstructive sleep apnea, sleep-disordered breathing, and excessive daytime sleepiness are rare. The reported prevalence rates of sleep-disordered breathing range from 4% to 46%. In polysomnographic analyses, Manley et al. observed obstructive or central sleep apnea in 43% of 7 patients. Excessive daytime sleepiness was detected in 43% of 28 patients [86].

Using MSLT, Crabtree et al. found a 5.7% rate of sleep-disordered breathing in a group of 68 CP patients of whom 82% had excessive daytime sleepiness [88]. Other polysomnography studies observed a prevalence rate of obstructive sleep apnea ranging from 4.0 to 5.8% [50,80,91]. The prevalence rate of obstructive sleep apnea syndrome was 46% in a study on obese or overweight patients with CP. As BMI was not correlated with apnea–hypopnea index, it can be speculated that severe hypothalamic obesity alone was not a sufficient explanation for the high rate of sleep apnea in CP patients. Furthermore, patients with CP showed more daytime sleepiness when compared with weight-matched controls (71% vs. 17%, by ESS). However, an association between ESS and apnea–hypopnea index was not observed [87], suggesting other risk factors for excessive daytime sleepiness. In a cross-sectional study, O’Gorman et al. observed a higher rate of sleep-disordered breathing (including central and obstructive sleep apneas) in CP patients when compared with a BMI-matched control group. The authors conclude that the hypothalamus plays an important and direct role for respiratory activity [106].

### 3.6. Disorders of Circadian Rhythms

Hypothalamic lesions can cause alterations of the sleep–wake rhythm. When compared to healthy controls, CP patients display a reduced rate of REM sleep, decreased sleep efficiency, and increased rates of nighttime awakening [107,108]. In actigraphy studies, a trend towards increased fragmentation of circadian rhythm with increased sleep-onset latency, increased nocturnal activity, irregular bedtime, increased daytime rest episodes with decreased mean 24 h plasma melatonin concentrations, and reduced mean nocturnal melatonin concentrations have been observed [35,44,107]. Other studies reported that decreased nocturnal melatonin concentrations are associated with excessive daytime sleepiness [35,59], reduced sleep duration, and impaired sleep efficiency [59]. Pickering et al. differentiated three distinct circadian profiles of melatonin secretion: normal, phase-shifted peak, and absent midnight peak. Patients without detectable midnight peak of melatonin secretion were the only ones presenting with reduced sleep quality, excessive daytime sleepiness, and mental and general fatigue [59]. Pickering et al. concluded that melatonin concentration in saliva could be a biomarker for dysregulation of circadian rhythms, and assessment of melatonin concentrations in saliva may help to diagnose disorders of circadian sleep–wake rhythms. With regard to the potential impact of cortisol on sleep, a few studies analyzed cortisol concentrations in saliva. The results were controversial. Müller et al. did not observe differences in daytime salivary cortisol levels when comparing CP patient with controls [35]. On the other hand, Pickering et al. reported on increased salivary cortisol levels, especially in the evening. Additionally, increased midnight cortisol concentrations were positively correlated with the frequency of waking during the night and reduced total time of sleep [59].

### 3.7. Fatigue

Beckhaus et al. [109] assessed the relation between hypothalamic syndrome and fatigue symptoms in CP patients. Hypothalamic syndrome was diagnosed in 25 of 41 (61%) patients with childhood-onset CP, in whom after adjustment for gender and age at study, increased scores of the physical domain (*β* = 3.39 [95% CI: 1.18–5.60]) and sum MFI-20 (*β* = 11.42 [95% CI: 2.06–20.79]) domain were observed, when compared with CP patients without hypothalamic syndrome. When compared with reference values of the general population, increased scores of each fatigue domain were found for all patients. A total of 6 of 25 (24%) patients with hypothalamic syndrome reported excessive daytime sleepiness. The observed fatigue scores were increased in CP patients suffering from hypothalamic syndrome, regardless of ESS findings. No significant correlation was found between fatigue and daytime sleepiness. Increased levels of overall physical fatigue were found in CP patients with hypothalamic syndrome (Figure 3). The authors concluded that in clinical practice, it is important to distinguish between fatigue and daytime sleepiness and to identify patients with or at risk for the development of hypothalamic syndrome.

### 3.8. Treatment of Sleep Disturbances

In patients with childhood-onset CP, the initial therapeutic approaches for sleep disturbances are comparable with those of the general population and include psychological and behavioral treatment [73,82,110,111,112,113]. A possible target for sleep interventions to improve sleep is increasing melatonin serum concentrations by melatonin substitution medication. Müller et al. reported on reduced melatonin concentration in saliva of CP patients, who complained about more frequent waking during the night. In a study including ten patients, melatonin medication was associated with improvements of physical activity and daytime sleepiness. However, data on the duration of melatonin substitution and outcome parameters on weight and sleep were not reported [35]. According to published guidelines, [114] melatonin medication can be considered in the pediatric age group for patients with dysregulation of circadian rhythms and hypothalamic obesity to improve excessive daytime sleepiness.

Furthermore, another potential target for sleep interventions is excessive daytime sleepiness. Two studies analyzed the association between sleep disturbances and effects on weight development. Ismail et al. observed that dextroamphetamine medication reduced daytime sleepiness in all patients with hypothalamic syndrome [115]. Morgenthaler et al. did not detect a significant effect of fenfluramine and fluoxetine medication on excessive daytime sleepiness or disturbances of sleep–wake cycles. According to published guidelines, medication with combinations of stimulating agents (amphetamines, methylphenidate, or modafinil) may be effective for treatment of narcolepsy and excessive daytime sleepiness in adults [116]. In four of five patients with CP, Crowley et al. [87] reported that modafinil medication resulted in an improvement of excessive daytime sleepiness. In another study on seven patients with CNS tumors, Rosen et al. [117] observed reduced sleepiness after medication with central stimulating agents. In a case report on three pediatric patients with suprasellar CNS tumors and narcolepsy, Marcus et al. [74] observed that modafinil—alone or in combination with dextroamphetamine—significantly improved excessive daytime sleepiness. Khan et al. [70] reported significant reductions of excessive daytime sleepiness after stimulant medication in thirty-seven survivors of CNS tumors (sixteen patients with CP). Following published guidelines [116], central stimulants like methylphenidate, dextroamphetamine, and modafinil can be considered for treatment of excessive daytime sleepiness in pediatric and adult age groups of patients with CP.

Obstructive sleep apnea syndrome represents a third target for sleep intervention. Guidelines for diagnostic and treatment options are published for pediatric and adult age groups [74,81]. In obese patients with CP, a prevalence rate of ~46%, for obstructive sleep apnea syndrome was published. In the general population, the prevalence rate for obstructive sleep apnea syndrome ranged from 0% to 61% in BMI-matched controls [87,106]. In a BMI-matched controlled trial, O’Gorman et al. [106] found higher apnea–hypopnea index scores, higher frequency of obstructive episodes, and reduced oxygen saturation during both non-rapid eye movement and rapid eye movement sleep in adolescent CP patients when compared with BMI-matched controls. In another matched–controlled study, Crowley et al. [87] could not detect a higher apnea–hypopnea index or any associations between BMI and apnea–hypopnea index in patients with CP. Based on their observations, the authors speculated that there should be other drivers for obstructive sleep apnea syndrome besides obesity.

Analyzing twelve CP cases with obstructive sleep apnea syndrome, Crowley et al. [87] found positive therapeutic effects of continuous positive airway pressure (CPAP) therapy on excessive daytime sleepiness in eight CP patients.

## 4. Summary of Recommendations for Management of Sleep Disturbances

Psychological and behavioral diagnostics and therapeutic interventions should be taken into consideration with the aim of improving the quality of sleep and excessive daytime sleepiness in CP patients with sleep disturbances (guidelines: [73,82,110,111]).Melatonin substitution should be initiated in patients with CP and sleep disturbances to reduce excessive daytime sleepiness (guideline: [114]).Central stimulants, for instance dextroamphetamine, modafinil, and methylphenidate, should be considered to reduce excessive daytime sleepiness in CP patients (guideline: [116]).Diagnostics for obstructive sleep apnea syndrome can be initiated with a low threshold. Therapeutic interventions such as CPAP are recommended for patients with CP and hypothalamic obesity (guidelines: [74,81]) (Table 1 and Table 2).

## 5. Future Perspectives in Management of Sleep Disturbances

Management of sleep disturbances in patients with CP is currently not very well explored. Multicenter studies on the correlation between hypothalamic damage scores and sleep disorders are still missing and could add valuable insights on the pathophysiological context. In CP patients with hypothalamic syndrome, impaired melatonin production can be associated with excessive daytime sleepiness [36]. However, published reports on reduced melatonin concentrations are contradictory [35,59,107,126]. It is speculated that various patterns of melatonin secretion are detectable in patients with CP. In patients suffering from suprasellar tumors including CP, the missing of a melatonin peak or phase-shifted dysregulation of melatonin peak concentrations are a well-known observation [59,60,107]. Future cohort studies are warranted and should analyze 24-h patterns of melatonin secretion for improved prediction, i.e., which patients would benefit from melatonin medication. Furthermore, prospective studies on the long-term effect of melatonin replacement therapy on sleep architecture are warranted.

In CP patients, clinical manifestations of secondary narcolepsy are diverse: SOREMPs, cataplexy, and reduced CSF concentrations of orexin can be detected [79,105,127,128]. Secondary narcolepsy might be underdiagnosed due to heterogeneous presentation and required testing modalities. Future studies with a large cohort size are warranted to provide a better knowledge of prevalence rate and clinical manifestations of narcolepsy in CP patients.

Melatonin receptor agonists, for instance agomelatine, ramelteon, tasimelteon, and circadin (a prolonged release melatonin agonist) are new pharmaceutical agents for the treatment of sleep disturbances that may represent a treatment option for CP patients [129,130,131,132,133,134]. Promising pharmaceutical agents for medical therapy of obstructive sleep apnea syndrome include dronabinol, a non-selective cannabinoid type I and II receptor agonist [135].

Further research on diagnostic criteria and specific clinical manifestations of hypothalamic syndrome in different diseases is warranted [123].

Faquih et al. recently reported that a metabolomic profile for excessive daytime sleepiness is characterized by endogenous and dietary metabolites within the steroid hormone biosynthesis pathway, with some pathways that differ by sex. These pathways need further investigation for understanding the broad spectrum of causes or consequences of excessive daytime sleepiness and related sleep disorders [136].

## 6. Conclusions

Sleep disturbances are frequently observed in CP patients with hypothalamic syndrome. The broad spectrum of pathogenic factors includes hypothalamic lesions, lesions of the suprachiasmatic nuclei, reduced melatonin concentrations, hypocretin deficit, and hypothalamic syndrome including morbid obesity. Furthermore, sleep disturbances in CP patients are characterized by various clinical manifestations such as secondary narcolepsy, excessive daytime sleepiness, dysregulation of circadian rhythms, disorders of sleep–wake transition, and sleep-disordered breathing. Based on the variety of these conditions, it can be challenging to come to a diagnosis. Misdiagnosis frequently leads to inadequate treatment. Appropriate management of these disorders may result in improvements in overall health and quality of survival in patients with CP.

## Figures and Tables

**Figure 1 biomedicines-13-02356-f001:**
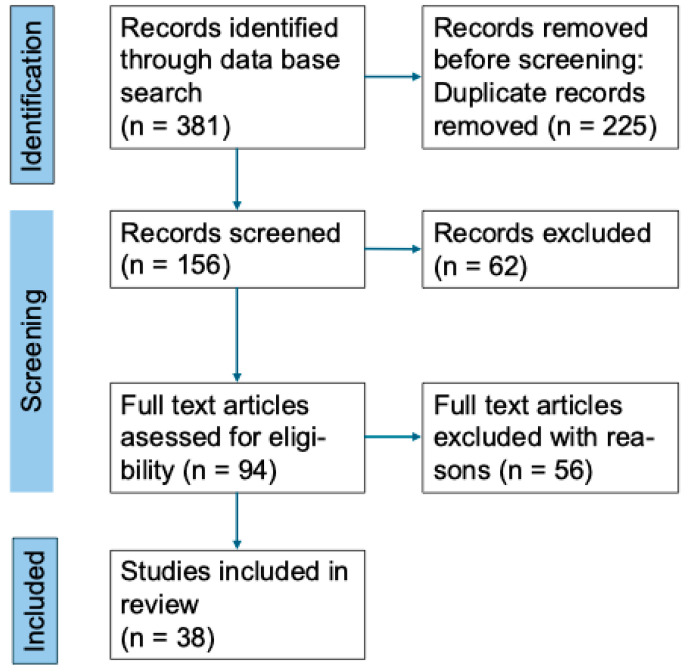
Preferred Reporting Items for Systematic reviews and Meta-Analyses (PRISMA) flow diagram of the literature review. *n*, patient number.

**Figure 2 biomedicines-13-02356-f002:**
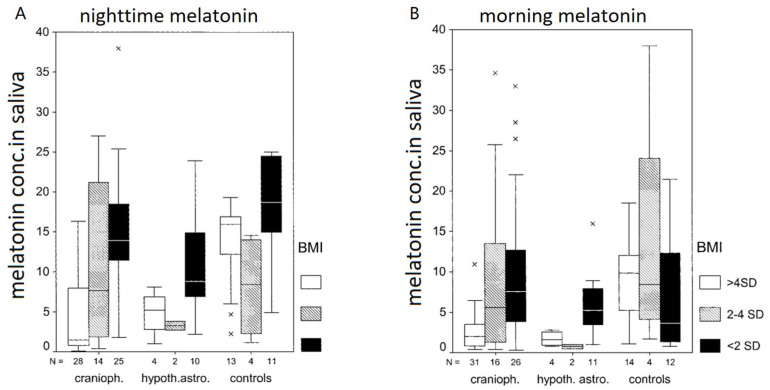
Salivary melatonin concentrations at nighttime (**A**) and in the morning (**B**) in patients with childhood craniopharyngioma, hypothalamic pilocytic astrocytoma, and controls in relation to the degree of obesity (body mass index (BMI) < 2 SD (filled black boxes), BMI 2–4 SD (hatched gray boxes], or BMI ≥ 4 SD (open boxes)). The horizontal line in the middle of the box depicts the median. Edges of the box mark the 25th and 75th percentile. Whiskers indicate the range of values that fall within 1.5 box lengths. Values more than 1.5 box length from the 25th and 75th percentiles are marked by an asterisk. (Reproduced from Müller et al., J Clin Endocrinol Metab, 2002 [36], with the kind permission of Endocrine Press). Abbreviations: cranioph., craniopharyngioma; hypoth. astro., hypothalamic astrocytoma; conc., concentration.

**Figure 3 biomedicines-13-02356-f003:**
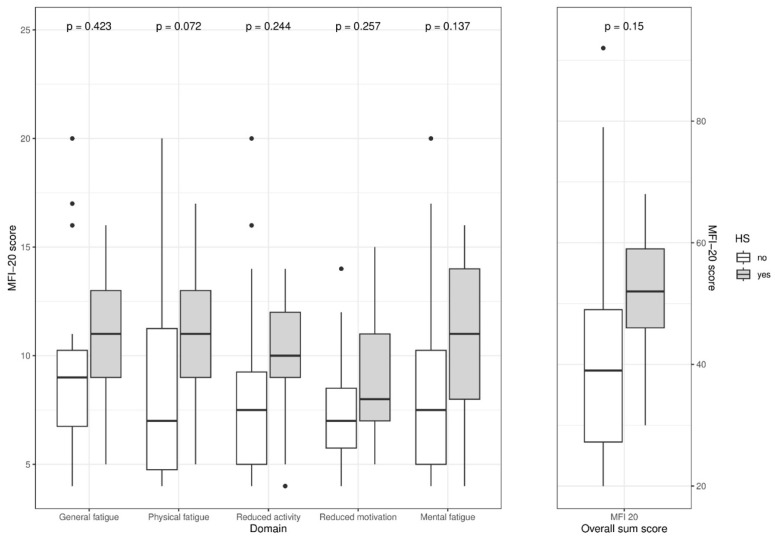
Boxplots showing multidimensional fatigue inventory (MFI)-20 scores of different domains (**left**) and overall sum score (**right**) in patients without (*n* = 16; white boxes) and with hypothalamic syndrome (HS) (*n* = 25; gray boxes). The horizontal line in the middle of the box depicts the median. The top and bottom edges of the box respectively mark the 25th and 75th percentiles. Whiskers indicate the range of values that fall within 1.5 box lengths. Outliers are indicated as dots. *p* values are retrieved from Student’s t-test. (Reproduced from Beckhaus et al., EJC Pediatric Oncology, 2024, [109] (with kind permission of Elsevier).

**Table 1 biomedicines-13-02356-t001:** Characteristics of studies included in this review.

CP pts/ Total pts	Age at Study	CO CP/ Total CP	Study Design	Methods, Instruments	Sleep Disturbances	Authors/Year of Publication
2/2	19, 12	2/2	Case series	MSLT, PSG	Patient 1: SDB, narcolepsy Patient 2: SDB, parasomnia	Cordani et al., 2021 [76]
70/70	6–20	70/70	Cross-sectional	ESS	EDS by M-ESS: 28.8%; EDS by MSLT: 81.8%; SDB: 5.7%	Crabtree et al., 2019 [88]
28/28	19–67	7/28	Case-control	Questionnaire	EDS by ESS: 25% sleep obstructive apnea: 46%	Crowley et al., 2011 [87]
4/7	17	3/4	Case series	MSLT, PSG	1 patient: EDS, improvement with dextroamphetamine	Denzer et al., 2019 [118]
13/13	17–76	1/13	Cross-sectional	NA	Presurgical: 20%, postsurgical: 8% sleep problems (n.s.)	Honegger et al., 1998 [119]
9/12	♂: med. 20 ♀: med. 15	9/9	Case series	MSLT, ESS, PSG	EDS, improvement with dexamphetamine: 8/12	Ismail et al., 2006 [115]
25/25	1–15	25/25	Cross sectional	ESS	Sleep disorders (n.s.): 12%	Kalapurakal et al., 2003 [120]
1/1	5–10	1/1	Case report	Actigraphy	Disturbed sleep pattern (frequent falling asleep, reversal sleep rhythm)	Killeffer et al., 1970 [83]
84/84	10.3 ± 4.3	84/84	Cross-sectional	ESS	Correlation btw. hypothalamic involvement and BMI/EDS	Klages et al., 2021 [50]
3/3	15–22	3/3	Cross-sectional	MSLT, PSG	Nighttime activity, inappropriate daytime episodes of rest	Lipton et al., 2009 [107]
3/10	6–16	3/3	Retrospective	ESS	Narcolepsy type 2	Madan et al., 2021 [102]
98/98	3–20	98/98	Cross-sectional	Questionnaire	EDS: 80%; hypersomnia: 45%; narcolepsy: 35%; OSA: 5%	Mandrell et al., 2020 [80]
28/28	10–32	28/28	Retrospective	MSLT, PSG	EDS: 43% (12/28); obstructive sleep apnea: 43% (3/7)	Manley et al., 2012 [86]
1/3	5	1/1	Case series	NA	Secondary narcolepsy	Marcus et al., 2002 [79]
79/79	3.5–33.2	79/79	Cross-sectional	MSLT, ESS, PSG	EDS: 35% (42% of severely obese) decreased nocturnal salivary melatonin levels	Müller et al., 2002 [36]
79/79	6–33.2	79/79	Cross-sectional	MSLT, PSG	EDS; reduced nocturnal melatonin levels	Müller et al., 2006 [35]
50/50	3–20	50/50	Cross-sectional	ESS	Hypersomnia: 50%	Niel et al., 2020 [91]
78/78	6–20	78/78	Cross-sectional	Actigraphy	EDS: 82%	Niel et al. 2021 [47]
15/15	10–21	15/15	Cross-sectional	ESS	SDB (AHI higher than controls)	O’Gorman et al., 2010 [106]
10/10	7.1–22.9	10/10	Cross-sectional	MSLT, PSG	Decreased rates of REM sleep, low sleep efficiency	Palm et al., 1992 [108]
15/15	18–70	4/15	Case-control	ESS	EDS, reduced sleep time and low midnight melatonin	Pickering et al., 2014 [59]
7/7	21–68	1/7	Case-control	Questionnaire	Hypersomnia: 57%	Pickering et al., 2017 [89]
21/21	<16	21/21	Cross-sectional	MSLT, PSG	EDS: 29%	Poretti et al., 2004 [84]
41/41	1–59	~ 50%	Retrospective/prospective	NA	Preoperative sleep disorders (n.s.): 15%	Ramanbhavana et al., 2019 [121]
1/1	19	1/1	Case report	MSLT, ESS, PSG	Secondary narcolepsy	Sakuta et al., 2012 [105]
1/1	29	1/1	Case report	MSLT, PSG	OSAS	Schultes et al., 2009 [122]
3/5	11–19	3/3	Cross-sectional	ESS	EDS	Snow et al., 2002 [90]
1/1	11	1/1	Case report	Actigraphy	Secondary narcolepsy	Tachibana et al., 2005 [78]
27/27	27–80	8/27	Case-control	ESS	EDS: 33%	Van der Klaauw et al., 2008 [85]
80/80	2–20	80/80	Cross-sectional	MSLT, PSG	Poor sleep	Witcraft et al., 2022 [44]
131/131	9–20	32/131	Cross-sectional	ESS	Worse EDS in bilateral-HI group	Yang et al., 2020 [93]
41/41	22 (13–45)	41/41	Cross-sectional	ESS, MFI-20	No correlation btw. fatigue and EDS, physical + overall fatigue high in pts. with HI	Beckhaus et al., 2024 [109]
119/119	22 (14–42)	119/119	Cross-sectional	ESS, PedQol	Negative correlation btw. ESS and QoL, EDS with posterior HI	Mann-Markutzyk et al., 2025 [94]
88/336	14 (6–29)	88/88	Retrospective cohort study	Hypothalamic score [123]	Mild sleep disorder: 14% Severe sleep disorder: 10%	Van Roessel et al., 2025 [13]
35/425	67 (16%) > 18 yrs	35/35	Retrospective cohort study	PedsQL, MFS	Fatigue not related to tumor location, worse in follow-up	Irestorm et al., 2024 [124]
54/54	37.1 ± 15.5	0/54	Retrospective cohort study	PSG, MSLT	Secondary narcolepsy:14% Hypersomnia: 26%	Dodet et al., 2023 [40]
109/109	40 (28–56)	0/109	Retrospective cohort study	PSQI	Sleep disturbance: 47/109 (43%)	Lin et al., 2023 [125]
62/62	11 ± 4.0	62/62	Cross-sectional	MSLT, PSG	EDS: 76%	Jacola et al., 2016 [92]

Abbreviations: ♂, male; ♀, female; CP, craniopharyngioma; CO, childhood-onset; pts, patients; yrs, years; Y, yes; NA, not available; PSG, polysomnography; MSLT, multiple sleep latency test; ESS, Epworth Sleepiness Scale; M-ESS, modified Epworth Sleepiness Scale; MEQ, Horne-Ostberg Morningness-Eveningness Questionnaire; EEG, electroencephalogram; SDB, sleep-disordered breathing; EDS, excessive daytime sleepiness; OSA, obstructive sleep apnea; OSAS, obstructive sleep apnea syndrome; BMI, body mass index; HI, hypothalamic injury; AHI, apnea–hypopnea index; MFS, multidimensional fatigue scale; QoL, quality of life; PSQI, Pittsburgh Sleep Quality Index; n.s., not specified. (modified from Cordani et al.) (2022) [37].

**Table 2 biomedicines-13-02356-t002:** Sleep disorders and therapeutic interventions in patients with CP.

Patient Cohort	Disturbance	Therapeutic Intervention	Effect/ Tolerability	Authors
10, adult obese patients	EDS	Melatonin (6 mg)	EDS improved (10/10 patients)	Müller et al., 2006 [35]
5, adult overweight/obese patients	EDS	Modafinil	EDS improved (4/5) (*one died before intervention)	Crowley et al., 2011 [87]
1, 5-year-old child	Secondary narcolepsy	Modafinil (200 mg) Methylphenidate (20 mg)	Improvement (No prolonged FU available)	Marcus et al., 2002 [79]
12, obese adolescent/young adult patients (9 with CP)	EDS	Dexamphetamine (5 mg twice daily)	EDS improved (8/12) Improved concentration and physical exercise tolerance (3/12) (1 discontinued for deteriorating health)	Ismail et al., 2006 [115]
1, 17-year-old obese boy	EDS	Dextroamphetamine	EDS improved	Denzer et al., 2019 [118]
7, adult overweight/obese patients	SDB EDS	NIV CPAP	EDS improved	Crowley et al., 2011 [87]
2, adolescent patients	SBD EDS	NIV CPAP	Resolution of SBD, EDS not improved	Snow et al., 2002 [90]
1	EDS	Correction of sleep hygiene	EDS improved	Manley et al., 2012 [86]

Abbreviations: EDS, excessive daytime sleepiness; SDB, sleep-disordered breathing; NIV, non-invasive ventilation; CPAP, continuous positive airway pressure. (modified from Cordani et al.) (2022) [37].

## Data Availability

The datasets generated and/or analyzed during the current study are available from the author upon reasonable request.

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
