# Peer review of "Sleep Disorders, Dysregulation of Circadian Rhythms, and Fatigue After Craniopharyngioma—A Narrative Review"

_biomedicines, 2025, doi:10.3390/biomedicines13102356_

Round 1
Reviewer 1 Report
Comments and Suggestions for Authors
This review provides a systematic summary of the pathophysiological mechanisms, clinical manifestations, assessment methods, and treatment strategies for sleep disorders, circadian rhythm dysregulation, and fatigue in patients with craniopharyngioma (CP). It holds significant clinical and scientific value. The author demonstrates a comprehensive grasp of the current state of research in this field, covering a wide time span of literature (1970–2025). The structure is clear, and the arguments are well-supported, making it a high-quality narrative review. However, there are still some suggestions for improvement:
- Lack of Methodological Description for Systematic Reviewing: It is advisable to briefly mention literature screening criteria (e.g., whether PRISMA flowchart was applicable), data extraction methods, and quality assessment processes to enhance methodological transparency and reproducibility.
- Some Data Presentations Could Be More Precise: For example, the stated prevalence range for excessive daytime sleepiness (EDS) is 25%–100%, which is very broad. It would be helpful to further analyze potential reasons for this variability (e.g., sample differences, assessment tools) or provide a weighted average if possible.
- Treatment Section Could Be Further Refined: While treatments like melatonin, central stimulants, and CPAP are mentioned, there is a lack of quantified summary of treatment effects (e.g., response rates, side effects). Consider adding a summary table aggregating the evidence level and recommendation strength for various treatment methods.
- Future Perspectives Could Be More Specific: Suggest proposing 2-3 concrete research directions or clinical trial design recommendations. For example, prospective studies on the effect of melatonin replacement therapy on sleep architecture, or multicenter studies on the correlation between hypothalamic damage scores and sleep disorders.
Author Response
Reviewer 1:
This review provides a systematic summary of the pathophysiological mechanisms, clinical manifestations, assessment methods, and treatment strategies for sleep disorders, circadian rhythm dysregulation, and fatigue in patients with craniopharyngioma (CP). It holds significant clinical and scientific value. The author demonstrates a comprehensive grasp of the current state of research in this field, covering a wide time span of literature (1970–2025). The structure is clear, and the arguments are well-supported, making it a high-quality narrative review. However, there are still some suggestions for improvement:
- Lack of Methodological Description for Systematic Reviewing: It is advisable to briefly mention literature screening criteria (e.g., whether PRISMA flowchart was applicable), data extraction methods, and quality assessment processes to enhance methodological transparency and reproducibility.
We are grateful for the reviewer’s comment and suggestion. We have added the following information to the Methods section of our revise manuscript:
Literature search strategy
- A comprehensive electronic literature search was conducted in PubMed/MEDLINE and Scopus databases in order to find relevant English-written published articles. The articles were identified using the following keywords: a) “craniopharyngioma” and b) “sleep,” “sleep disorders,” “sleep-related breathing disorders,” “sleep-disordered breathing,” “excessive daytime sleepiness,” “hypersomnolence,” “narcolepsy”, ”fatigue”, “circadian rhythm,” “melatonin,” “stimulant.”
Inclusion and Exclusion Criteria
- Only studies published in peer-reviewed journals were considered. The inclusion criteria, based on the PICOS approach were as follows: Participants (P): patients diagnosed with craniopharyngioma; Intervention (I): no restrictions; Comparison (C): no restrictions; Outcomes (O): sleep-related parameters; Study design (S): original studies. Exclusion criteria included: 1) non-English language publications; 2) animal studies or in vitro research; 3) studies involving participants with craniopharyngioma with additional central nervous system complications; 4) reviews, case report, thesis, and conference proceedings. In cases where multiple studies were published based on the same databases, only the study with the largest sample size was included in the analysis.
Data Extraction and Synthesis
- The data extracted from all studies in line with our research objectives included participants characteristics (i.e., age of the subjects, age of tumor onset, clinical characteristics), procedures used evaluate sleep disorders (subjective and/or objective investigations), type of sleep disturbance, any additional examinations (melatonin dosage), treatment and efficacy, if available.
Furthermore, the following PRISMA flow diagram of search results was added to our revised manuscript:
Figure 1: PRISMA flow diagram of the literature review.
- Some Data Presentations Could Be More Precise: For example, the stated prevalence range for excessive daytime sleepiness (EDS) is 25%–100%, which is very broad. It would be helpful to further analyze potential reasons for this variability (e.g., sample differences, assessment tools) or provide a weighted average if possible.
We are grateful for the reviewer’s comment and suggestion. We have already added the following information, which helps to understand the wide range of prevalence for excessive daytime sleepiness reported in the literature:
Lines 183-205: “Considering all reports regardless of diagnostic techniques used, excessive daytime sleepiness is recognized with a rate ranging from 25% to 100% of cases. Different cohort sizes and criteria for patient selection could explain the wide range of variability. Lower prevalence values for somnolence - ranging from 25% to 43% of cases - have been observed in analyses questionnaires. Poretti et al. observed excessive daytime sleepiness in 29% of patients with childhood-onset CP [84]. In an ESS study on 79 CP patients, Müller et al. reported excessive daytime sleepiness in 42% of severely obese and in 35% of less obese or normal weight CP patients. Van der Klaauw et al. observed hypersomnolence in 33% of CP patients [85]. The authors compared healthy controls with CP patients and patients with non-functioning pituitary macroadenomas. They observed that patients with CP or non-functioning pituitary macroadenomas have excessive daytime sleepiness despite normal nocturnal sleep patterns [85]. Manley et al. observed excessive daytime sleepiness in 43% of patients with childhood-onset CP. [86]. An excessive daytime sleepiness (ESS values >10) prevalence rate of 25% was reported by Crowley et al. [87]. Using polysomnography and MSLT as the standard for objective assessment of excessive daytime sleepiness, the prevalence rate of excessive daytime sleepiness in CP patients appears to be higher, ranging from 50% to 100% of cases. Based on mean sleep latency values <10 min in MSLT, Crabtree et al. found that 82% of pediatric CP patents presented with excessive daytime sleepiness [88]. Using a modified ESS, the authors observed excessive daytime sleepiness in 29% of patients. The results lead to the speculation that CP patients often do not correctly report or recognize their sleep disorders. In polysomnography and MSLT after surgical resection of CP, Mandrell et al. observed excessive daytime sleepiness in 80% of patients with CP [80].“
- Treatment Section Could Be Further Refined: While treatments like melatonin, central stimulants, and CPAP are mentioned, there is a lack of quantified summary of treatment effects (e.g., response rates, side effects). Consider adding a summary table aggregating the evidence level and recommendation strength for various treatment methods.
We are grateful for the reviewer’s suggestion. We have added a new Table 2, which shows available data on therapeutic interventions and their effect. However, most data are based on very small cohorts and reliable data on response rates are mostly not available.
Table 2: Sleep disorders and therapeutic interventions in patients with CP.
Patient cohort |
Distur-bance |
Therapeutic intervention |
Effect / tolerability |
Authors / year of publication |
10, adult obese patients |
EDS |
Melatonin (6 mg) |
EDS improved |
Müller et al., 2006 [35] |
5, adult overweight/obese patients |
EDS |
Modafinil |
EDS improved (4/5) |
Crowley et al., 2011 [87] |
1, 5-year-old child |
Secondary nar-colepsy |
Modafinil (200 mg) |
Improvement |
Marcus et al., 2002 [79] |
12, obese adolescent/young adult patients (9 with CP) |
EDS |
Dexamphetamine |
EDS improved (8/12) |
Ismail et al., 2006 [115] |
1, 17-years old obese boy |
EDS |
Dextroamphetamine |
EDS improved |
Denzer et al., 2019 [128] |
7, adult overweight/ obese patients |
SDB |
NIV CPAP |
EDS improved |
Crowley et al., 2011 [87] |
2, adolescent patients |
SBD |
NIV CPAP |
Resolution of SBD, |
Snow et al., 2002 [90] |
1 |
EDS |
Correction of sleep quality |
EDS improved |
Manley et al., 2012 [86] |
Abbreviations: EDS, excessive daytime sleepiness; SDB, sleep-disordered breathing; NIV, non-invasive ventilation; CPAP, Continuous positive airway pressure.
- Future Perspectives Could Be More Specific: Suggest proposing 2-3 concrete research directions or clinical trial design recommendations. For example, prospective studies on the effect of melatonin replacement therapy on sleep architecture, or multicenter studies on the correlation between hypothalamic damage scores and sleep disorders.
We are grateful for the reviewer’s suggestion and have added concrete research directions to our revised manuscript:
- Lines 426-428: Multicenter studies on the correlation between hypothalamic damage scores and sleep disorders are still missing and could add valuable insights on the pathophysiological context.
- Lines 436-437: Furthermore, prospective studies on the long-term effect of melatonin replacement therapy on sleep architecture are warranted.
- Lines 450-454: Faquih et al. recently reported that a metabolomic profile for excessive daytime sleepiness is characterized by endogenous and dietary metabolites within the steroid hormone biosynthesis pathway, with some pathways that differ by sex. These pathways need further investigation for understanding the broad spectrum of causes or consequences of excessive daytime sleepiness and related sleep disorders [136].

Reviewer 2 Report
Comments and Suggestions for Authors
The submitted manuscript provides a narrative review on sleep disorders, circadian rhythm dysregulation, and fatigue following craniopharyngioma. While the topic is clinically relevant, the paper unfortunately does not meet the standards required for publication. The work suffers from significant methodological weaknesses, the literature search is inadequately described, lacks a systematic framework, and does not provide transparent inclusion or exclusion criteria. The narrative largely reiterates well-established pathophysiology and clinical associations without offering novel synthesis, critical appraisal, or new insights. Much of the content is repetitive, overly descriptive, and relies heavily on secondary sources rather than providing an evaluative discussion of evidence quality or limitations. Figures and tables reproduce previous work without adding substantive value. Furthermore, the conclusions are general and speculative, not clearly supported by a structured analysis of the cited literature. Given these limitations, the manuscript in its current form adds little to the existing body of knowledge and does not justify publication.
Author Response
Rebuttal - biomedicines-3859820: September 10, 2025
“Sleep disorders, dysregulation of circadian rhythms and fatigue
after craniopharyngioma — a narrative review”,
by Hermann L. Müller
Reviewer 2:
The submitted manuscript provides a narrative review on sleep disorders, circadian rhythm dysregulation, and fatigue following craniopharyngioma. While the topic is clinically relevant, the paper unfortunately does not meet the standards required for publication. The work suffers from significant methodological weaknesses, the literature search is inadequately described, …
Rebuttal: We are grateful for the reviewer’s comment and suggestion. We have added extensive information on methodology and our literature search including a PRISMA flow chart (Figure 1). We have added the following information to the Methods section of our revise manuscript:
Literature search strategy
- A comprehensive electronic literature search was conducted in PubMed/MEDLINE and Scopus databases in order to find relevant English-written published articles. The articles were identified using the following keywords: a) “craniopharyngioma” and b) “sleep,” “sleep disorders,” “sleep-related breathing disorders,” “sleep-disordered breathing,” “excessive daytime sleepiness,” “hypersomnolence,” “narcolepsy”, ”fatigue”, “circadian rhythm,” “melatonin,” “stimulant.”
Inclusion and Exclusion Criteria
- Only studies published in peer-reviewed journals were considered. The inclusion criteria, based on the PICOS approach were as follows: Participants (P): patients diagnosed with craniopharyngioma; Intervention (I): no restrictions; Comparison (C): no restrictions; Outcomes (O): sleep-related parameters; Study design (S): original studies. Exclusion criteria included: 1) non-English language publications; 2) animal studies or in vitro research; 3) studies involving participants with craniopharyngioma with additional central nervous system complications; 4) reviews, case report, thesis, and conference proceedings. In cases where multiple studies were published based on the same databases, only the study with the largest sample size was included in the analysis.
Data Extraction and Synthesis
- The data extracted from all studies in line with our research objectives included participants characteristics (i.e., age of the subjects, age of tumor onset, clinical characteristics), procedures used evaluate sleep disorders (subjective and/or objective investigations), type of sleep disturbance, any additional examinations (melatonin dosage), treatment and efficacy, if available.
Furthermore, a PRISMA flow diagram of search results was added to the revised manuscript:
… lacks a systematic framework and does not provide transparent inclusion or exclusion criteria.
Rebuttal: Inclusion and exclusion criteria were described in a distinct methodological section.
Inclusion and Exclusion Criteria
- Only studies published in peer-reviewed journals were considered. The inclusion criteria, based on the PICOS approach were as follows: Participants (P): patients diagnosed with craniopharyngioma; Intervention (I): no restrictions; Comparison (C): no restrictions; Outcomes (O): sleep-related parameters; Study design (S): original studies. Exclusion criteria included: 1) non-English language publications; 2) animal studies or in vitro research; 3) studies involving participants with craniopharyngioma with additional central nervous system complications; 4) reviews, case report, thesis, and conference proceedings. In cases where multiple studies were published based on the same databases, only the study with the largest sample size was included in the analysis.
The narrative largely reiterates well-established pathophysiology and clinical associations without offering novel synthesis, critical appraisal, or new insights.
Rebuttal: Especially “hypothalamic syndrome” is a new, recently described clinical manifestation in craniopharyngioma, which includes sleep disorders. In this new context, noval clinical associations are described in the review.
Table 2: Sleep disorders and therapeutic interventions in patients with CP.
Patient cohort |
Distur-bance |
Therapeutic intervention |
Effect / tolerability |
Authors / year of publication |
10, adult obese patients |
EDS |
Melatonin (6 mg) |
EDS improved |
Müller et al., 2006 [35] |
5, adult overweight/obese patients |
EDS |
Modafinil |
EDS improved (4/5) |
Crowley et al., 2011 [87] |
1, 5-year-old child |
Secondary nar-colepsy |
Modafinil (200 mg) |
Improvement |
Marcus et al., 2002 [79] |
12, obese adolescent/young adult patients (9 with CP) |
EDS |
Dexamphetamine |
EDS improved (8/12) |
Ismail et al., 2006 [115] |
1, 17-years old obese boy |
EDS |
Dextroamphetamine |
EDS improved |
Denzer et al., 2019 [128] |
7, adult overweight/ obese patients |
SDB |
NIV CPAP |
EDS improved |
Crowley et al., 2011 [87] |
2, adolescent patients |
SBD |
NIV CPAP |
Resolution of SBD, |
Snow et al., 2002 [90] |
1 |
EDS |
Correction of sleep quality |
EDS improved |
Manley et al., 2012 [86] |
Much of the content is repetitive, overly descriptive, and relies heavily on secondary sources rather than providing an evaluative discussion of evidence quality or limitations. Figures and tables reproduce previous work without adding substantive value.
Rebuttal: The two tables are summarizing the clinical information of papers published on the topic of the review. Important conclusions are drawn based on the literature review. We furthermore, designed a new table depicting therapeutic interventions and their effects and tolerability for therapy of sleep disorders in patients with craniopharyngioma.
see Table 2 above
Abbreviations: EDS, excessive daytime sleepiness; SDB, sleep-disordered breathing; NIV, non-invasive ventilation; CPAP, Continuous positive airway pressure.
Furthermore, the conclusions are general and speculative, not clearly supported by a structured analysis of the cited literature.
Rebuttal: In our revised manuscript, we have added practical and concrete conclusions and recommendations for future studies as recommended by the reviewer.
- Lines 426-428: Multicenter studies on the correlation between hypothalamic damage scores and sleep disorders are still missing and could add valuable insights on the pathophysiological context.
- Lines 436-437: Furthermore, prospective studies on the long-term effect of melatonin replacement therapy on sleep architecture are warranted.
- Lines 450-454: Faquih et al. recently reported that a metabolomic profile for excessive daytime sleepiness is characterized by endogenous and dietary metabolites within the steroid hormone biosynthesis pathway, with some pathways that differ by sex. These pathways need further investigation for understanding the broad spectrum of causes or consequences of excessive daytime sleepiness and related sleep disorders [136].
Given these limitations, the manuscript in its current form adds little to the existing body of knowledge and does not justify publication.
Rebuttal: We are grateful for the reviewer’s comments. Hopefully, I could improve the revised manuscript by following the reviewer’s suggestions and criticism.

Round 2
Reviewer 2 Report
Comments and Suggestions for Authors
The authors did a good job.